# Cannula Fracture during Transoral Endoscopic Thyroidectomy Vestibular Approach: Causes and Prevention

**DOI:** 10.3390/diagnostics12071566

**Published:** 2022-06-28

**Authors:** Tsung-Jung Liang, Shiuh-Inn Liu, I-Shu Chen

**Affiliations:** 1Division of General Surgery, Department of Surgery, Kaohsiung Veterans General Hospital, Kaohsiung 813414, Taiwan; s871114@ym.edu.tw (S.-I.L.); nugaticc@gmail.com (I.-S.C.); 2School of Medicine, National Yang Ming Chiao Tung University, Taipei 112304, Taiwan

**Keywords:** cannula fracture, trocar, transoral endoscopic thyroidectomy, vestibular approach, thyroid

## Abstract

In the transoral endoscopic thyroidectomy vestibular approach (TOETVA), three oral vestibular incisions are used to access the thyroid. This approach leaves no scar on the body surface; however, unexpected complications may occur. Three patients (two women, one man) underwent TOETVA using the standard three-port technique. Broken cannulas of the 12 mm central port were noted in all cases. All cannulas broke on the ventral side of the distal shaft. The fracture lines were 3–4 cm in length, with some fragments scattered throughout the operative field and oral cavity. The fractures were caused by compression against the mandible while tilting the cannula during surgical manipulation. Male sex, short stature, and protruding chin may be risk factors for cannula fracture in TOETVA. Measures should be taken to prevent this complication, particularly in high-risk patients.

## 1. Introduction

The transoral endoscopic thyroidectomy vestibular approach (TOETVA) is a novel technique with excellent cosmetic outcomes [1]. Compared with other remote-access thyroidectomies, such as those via the axilla or breast, the vestibular approach has the shortest distance to the neck and requires lesser skin-flap dissection [2,3]. Because this technique is minimal invasive, it has been quickly adopted worldwide since Dr. Anuwong published his first case series in 2016 [1,4].

Current evidence shows comparable surgical outcomes between transoral and conventional thyroidectomy [5]. However, using three incisions in the oral vestibule to access the thyroid might cause some complications that are unique to TOETVA and unseen in the conventional approach, such as mental nerve injury and mentalis muscle laceration [6,7]. Furthermore, the trocar is placed in a small confined space above the mandible, and complete breakdown of the trocar cannula into two parts with some scattered fragments has been reported [8]. The sharp edge of the broken cannula may injure the surrounding tissue. Moreover, small dislodged fragments may be dispersed widely and difficult to find. Needless to say, retained foreign objects are a serious threat to patient safety and may trigger a medical malpractice claim [9].

Herein, we present three cases of cannula fracture during TOETVA. We summarize the fracture pattern, possible mechanisms, and predisposing factors for cannula fractures. We also propose strategies for the prevention and management of cannula breakage.

## 2. Case Presentation

### 2.1. Case 1

A 57-year-old man (height, 166 cm; weight, 82 kg) presented with a 1.7 cm left thyroid nodule that was incidentally found during a health examination. Fine-needle aspiration cytology (FNAC) suggested that the nodule was a follicular neoplasm, and the patient was referred for surgical resection for a definitive diagnosis. He had no history of head and neck surgery.

The patient underwent transoral endoscopic left thyroidectomy. The operative procedure was similar to Dr. Anuwong’s technique with some refinements, as described in our previous publication [10,11]. One 12 mm trocar (ENDOPATH XCEL Trocar, Ethicon, Cincinnati, OH, USA), and two 5 mm trocars (VersaOne Optical Trocar, Medtronic, Minneapolis, MN, USA) were placed. A 30-degree 10 mm endoscope was inserted into the central trocar for inspection. The operative procedure was uneventful. However, a linear fracture of the distal part of the central port was found at the end of the surgery (Figure 1). The fracture line was located on the ventral side of the cannula, with a slight deviation to the left. After cannula removal, no bleeding was observed at the fracture site. The patient’s postoperative course was uneventful, and the final pathologic diagnosis revealed a non-invasive follicular thyroid neoplasm with papillary-like nuclear features.

### 2.2. Case 2

A 56-year-old woman (height, 154 cm; weight, 80 kg) was found to have a 1.9 cm left thyroid nodule during a health examination. FNAC of the lesion suggested papillary carcinoma. Preoperative computed tomography revealed no evidence of cervical lymph node metastases.

The patient underwent transoral endoscopic left thyroidectomy. After complete dissection of the thyroid gland, the central port was withdrawn to remove the surgical specimen, which was enclosed in an endo bag (Endo Pocket, UNIMAX, New Taipei, Taiwan). A broken cannula with a defect in the distal part was observed (Figure 2A). The cannula defect was on the ventral side, which was in contact with the mandible (Figure 2B). Missing fragments constituting the cannula defect were found in the operative field. Postoperatively, the patient recovered smoothly. Pathological examination revealed non-invasive follicular thyroid neoplasm with papillary-like nuclear features.

### 2.3. Case 3

A 45-year-old woman (height, 154 cm; weight, 45 kg) presented with progressive enlargement of a right thyroid goiter up to 4.4 cm, which caused tracheal deviation and compression. The patient underwent transoral endoscopic right thyroidectomy for symptom relief. Similar to Case 2, a broken cannula with a defect at the distal end was found when we extracted the 12 mm port to gain access for specimen retrieval (Figure 3A). The fracture lines were located on the ventral side of the cannula (Figure 3B). Three of the four dislodged fragments were retrieved from the operative field using a Maryland dissector. The last missing piece was found in the oral cavity. It lay on the gauze, which was placed in the mouth to absorb excessive saliva.

Patient characteristics and patterns of cannula fracture in all three cases are summarized in Table 1 and Table 2, respectively.

## 3. Discussion

To the best of our knowledge, this is the first case series focusing on the breakage of trocar cannulas during TOETVA. In our series, all cannulas broke at the distal end with similar fracture lengths (3–4 cm), and all fractures occurred on the ventral side where the cannulas were in direct contact with the mandible. The breakages only differed in the severity, from a single linear fracture to multiple fractures with two to four displaced fragments (Table 2). Hong et al. reported complete cannula rupture into two parts, which represents the most severe damage [8]. The similar characteristics of the broken cannulas imply that all ruptures were caused by the same mechanism.

We speculate that the cannulas fractured due to compression against the mandible (Figure 4). During the operation, the observation port must be tilted down to inspect the operative field. This action applies a force against the mandible that results in an equal counterforce to the cannula. Because the cannula is confined to the small and tight premandibular space with a limited range of motion, the accumulated compression pressure leads to cannula breakage. This theory can explain why all fractures occurred on the ventral side and in the distal shaft of the cannula, which is the fulcrum of the cannula on the mandible.

Because men tend to have prominent jaws and dense premandibular soft tissue, the cannula is encased more tightly, making it more susceptible to breakage during surgical manipulation, as in case 1 (Table 2) [12]. Patients 2 and 3 were relatively short (both women were 154 cm in height), which translated into proportionally smaller oral vestibules and chins. It is more difficult for a small chin to accommodate a standard 12 mm port; thus, a small chin may be a predisposing factor for cannula fracture. In addition, Case 3 had a protruding chin and the distance between the jaw and neck (length of jaw) was greater, even with the neck extended (Figure 4) [13]. Greater force had to be applied to tilt the central cannula, which introduced the endoscope, downwards to visualize the operative field properly. This excessive pressure increased the risk of cannula fractures. This may also explain why patients with a prominent mandibular jawline have more postoperative swelling and perceive more pain after TOETVA [13].

Table 3 summarizes the proposed strategies for preventing cannula fractures. Trocars with smaller diameters (e.g., 5 mm or 11 mm) can be used in patients with small chins to decrease passive compression caused by the surrounding tissue and mandible. Cohen et al. suggest using a 5 mm central trocar to replace the 12 mm one in pediatric patients [14]. Indeed, the premandibular space can be dilated to relieve compression, facilitate trocar movement, and remove specimens. However, routine space dilatation is not recommended because it may injure the surrounding mental nerve, which provides sensory innervation to the lower lip and chin, and/or the mentalis muscle, which elevates the lower lip and allows it to pout [7,11].

The movement of the endoscope should be gentle so that the cannula is not forcibly compressed against the mandible. Using a smaller 5 mm endoscope is another option to mitigate compression [15]. Further, the cannula can be retracted back to the lower edge of the chin [16]. Thus, the distal cannula would not compress against the mandible when the endoscope is tilted down to inspect the operative field. This adjustment of port position also enhances the endoscopic view of the superior thyroid pole [16]. Finally, we could replace the plastic trocar with a metal trocar, which is more fracture-resistant and reusable [8].

Using robotic systems with flexible scopes may be another solution for this problem [17]. Compared to the standard rigid endoscope, a flexible scope can navigate the entire operative field because the direction of the scope tip can be adjusted; therefore, the main scope shaft does not need to compress the mandible to gain access for better visualization [17]. For example, the da Vinci single port robotic surgery system (Intuitive Surgical, Sunnyvale, CA, USA) has been used in TOETVA procedures with some promising results [18]. Moreover, newer surgical robots are equipped with force sensor and force control algorithms to provide haptic feedback based on the stiffness of the tissue being manipulated [19]. This can prevent the application of excessive force and further improve patient safety.

Nevertheless, if the cannula breaks, a thorough inspection of the operative field should be performed to check for lacerations in the region of fracture and to collect all dislodged fragments. The oral cavity is another potential space where fragments may be deposited and should be checked. The fragments should be retrieved one by one and no missing pieces should be left in the patient’s body.

In our institute, the incidence rate of cannula fracture during TOETVA is 2.3% (3/132), which is comparable to 3.7% (3/82) reported by Hong et al. [8]. By following the prevention strategies in Table 3, the incidence rate might decline in future.

## 4. Conclusions

In summary, we reported three cases of broken cannulas during TOETVA. All breakages occurred on the ventral side of the distal cannula and were associated with mandibular compression. Male sex, short stature, or a protruding chin may be risk factors for cannula fracture. Using a smaller, break-resistant cannula, gentle manipulation, and pulling the cannula back to the lower edge of the chin might reduce the risk of breakage. Once a cannula breaks, thorough inspection of the operative field to retrieve all fragments is of utmost importance.

## Figures and Tables

**Figure 1 diagnostics-12-01566-f001:**
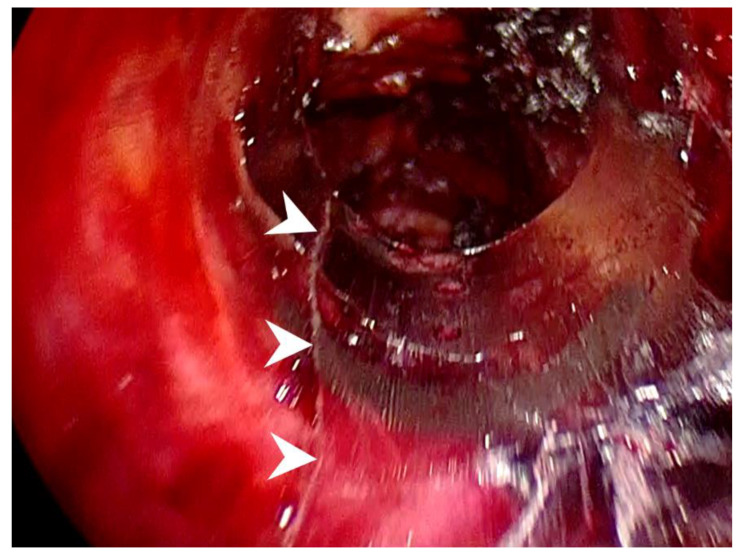
Intraoperative view of Case 1 demonstrating a linear fracture on the left ventral side of the cannula (**arrowheads**).

**Figure 2 diagnostics-12-01566-f002:**
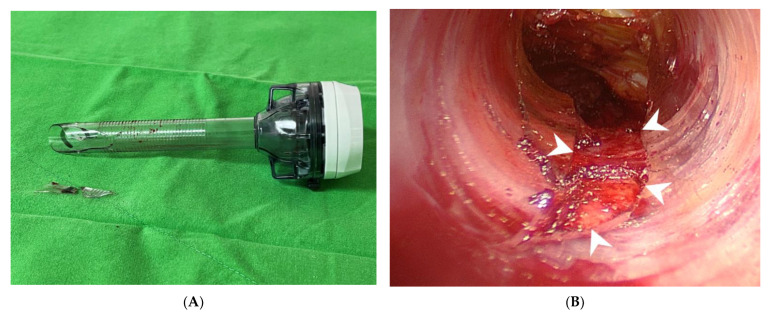
The broken cannula in Case 2. (**A**) The fracture is in the distal end of the cannula. Two fragments were retrieved. (**B**) Intraoperative view shows a cannula fracture with a defect on the ventral side of the cannula (**arrowheads**).

**Figure 3 diagnostics-12-01566-f003:**
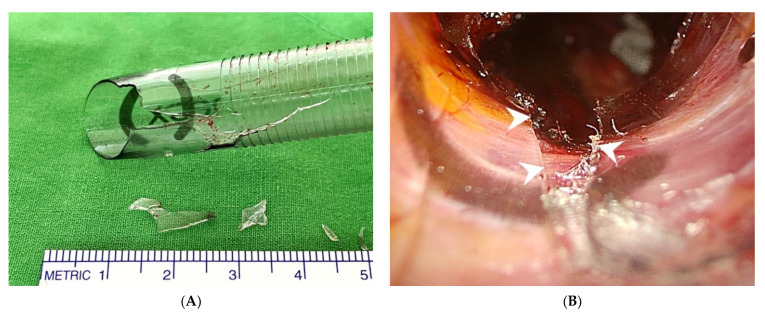
The broken cannula in Case 3. (**A**) The fractured part is at the distal end of cannula. Four fragments were retrieved. (**B**) Intraoperative view shows a cannula fracture with a defect on the ventral side of the cannula (**arrowheads**).

**Figure 4 diagnostics-12-01566-f004:**
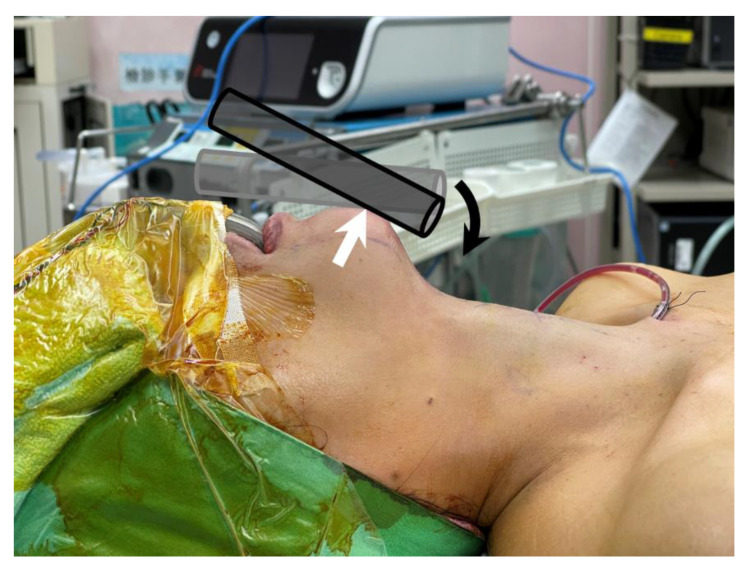
Proposed mechanism for cannula fracture. During the surgery, the observation port (**black cylinder**) is tilted down (**black arrow**) to inspect the operative field. This action applies a force against the mandible, which results in an equal counterforce (**white arrow**) to the cannula that leads to cannula breakage.

**Table 1 diagnostics-12-01566-t001:** Patient characteristics.

Case	Age	Sex	Height (cm)	Weight (kg)	BMI (kg/m^2^)	Diagnosis	Nodule Size (cm)	Laterality	Operative Time (min)
1	57	M	166	82	29.8	NIFTP	1.7	Left	175
2	56	F	154	68	28.6	NIFTP	1.9	Left	229
3	45	F	154	45	19.1	FA	4.4	Right	230

Abbreviations: BMI, body mass index; F, female; FA, follicular adenoma; M, male; NIFTP, non-invasive follicular thyroid neoplasm with papillary like nuclear features.

**Table 2 diagnostics-12-01566-t002:** Patterns of cannula fracture.

Case	Location	Side	Fracture Length (cm)	Fracture Type	Predisposing Factor
1	Distal *	Ventral ^†^	4	Single linear fracture without fragmentation	Tight pre-mandibular soft tissue
2	Distal	Ventral	3	Multiple fractures with 2 dislodged fragments	Short stature
3	Distal	Ventral	4	Multiple fractures with 4 dislodged fragments	Short stature, protruding chin

* distal end of the trocar cannula. ^†^ the side that contacts the mandible during the surgery.

**Table 3 diagnostics-12-01566-t003:** Strategies for preventing cannula fracture.

Mechanism	Strategy
Decrease passive compression	Small trocar (e.g., 5 mm or 11 mm in diameter) Dilate the premandibular space
Avoid active compression	Avoid tilting the cannula rigorously Use small endoscope Retract the cannula back to the lower edge of the chin
Increase resistance to break	Use break-resistant cannula or metal cannula

## Data Availability

All data generated or analyzed during this study are included in this published article.

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
