# Peer review of "Cannula Fracture during Transoral Endoscopic Thyroidectomy Vestibular Approach: Causes and Prevention"

_diagnostics, 2022, doi:10.3390/diagnostics12071566_

Round 1

Reviewer 1 Report

The article describes minimal invasive transoral endoscopic thyroidectomy from vestibular approach (TOETVA), a way to leave no scar on the body-skin surface. The writers perceived 12mm central port canula breaks as intraoperative complication in 3 cases (3/132, 2.3%). The paper aims to highlight the technical issues and looking for solutions how to prevent these.

They identified scattered fragmentation of the canula on the distal shaft part, on ventral side. All fragments were successfully retrieved from the operative field. Explanation was given as its compression against the mandible during surgical manipulation. Risk factors of these complications are male sex, protruding chin with short stature.

The paper clearly draws attention on the severe injuries broken fragments could cause on the surrounding tissues and the difficulties to find small dispersed fragments. Discussion completely included the possible outcomes of cannula fractures and explained the mechanical background of those. Different strategies were presented aiming for prevention, as using smaller trocars, avoiding rigorous tilting, retracting cannula back to chin or using metal, break-resistant cannulas.

Authors should provide some outlook to alternative robotic solutions for the TOTEVA procedures (see e.g., https://pubmed.ncbi.nlm.nih.gov/22020888/), and also discuss the possibility for measuring forces e.g., Haidegger, Tamás, Balázs Benyó, Levente Kovács, and Zoltán Benyó. "Force sensing and force control for surgical robots." IFAC Proceedings Volumes 42, no. 12 (2009): 401-406.

Author Response

The article describes minimal invasive transoral endoscopic thyroidectomy from vestibular approach (TOETVA), a way to leave no scar on the body-skin surface. The writers perceived 12mm central port canula breaks as intraoperative complication in 3 cases (3/132, 2.3%). The paper aims to highlight the technical issues and looking for solutions how to prevent these.

They identified scattered fragmentation of the canula on the distal shaft part, on ventral side. All fragments were successfully retrieved from the operative field. Explanation was given as its compression against the mandible during surgical manipulation. Risk factors of these complications are male sex, protruding chin with short stature.

The paper clearly draws attention on the severe injuries broken fragments could cause on the surrounding tissues and the difficulties to find small dispersed fragments. Discussion completely included the possible outcomes of cannula fractures and explained the mechanical background of those. Different strategies were presented aiming for prevention, as using smaller trocars, avoiding rigorous tilting, retracting cannula back to chin or using metal, break-resistant cannulas.

Authors should provide some outlook to alternative robotic solutions for the TOTEVA procedures (see e.g., https://pubmed.ncbi.nlm.nih.gov/22020888/), and also discuss the possibility for measuring forces e.g., Haidegger, Tamás, Balázs Benyó, Levente Kovács, and Zoltán Benyó. "Force sensing and force control for surgical robots." IFAC Proceedings Volumes 42, no. 12 (2009): 401-406.

Response:

Thank you for your comments.

Accordingly, we have added a paragraph regarding the alternative robotic solutions for TOETVA in the “Discussion” section (page 6, lines 153-162) as follows:

“Using robotic systems with flexible scopes may be another solution for this problem. Compared to the standard rigid endoscope, a flexible scope can navigate the entire operative field because the direction of the scope tip can be adjusted; therefore, the main scope shaft does not need to compress the mandible to gain access for better visualization. For example, the da Vinci single port robotic surgery system (Intuitive Surgical, Sunnyvale, CA) has been used in TOETVA procedures with some promising results. Moreover, newer surgical robots are equipped with force sensor and force control algorithms to provide haptic feedback based on the stiffness of the tissue being manipulated. This can prevent the application of an excessive force and further improve patient safety.”

Reviewer 2 Report

The authors discuss a highly needed and innovative topic herein, which represents a critical advancement in minimally invasive thyroidectomy procedure safety. 

Author Response

The authors discuss a highly needed and innovative topic herein, which represents a critical advancement in minimally invasive thyroidectomy procedure safety.

Response:

Thank you for your encouraging comments.